# Perinatal Management in a Pregnant Woman with Ureteropelvic Junction Obstruction: Case Report and Literature Review

**DOI:** 10.3390/diagnostics12040913

**Published:** 2022-04-06

**Authors:** Daisuke Tamura, Shintaro Narita, Misa Yamauchi, Rina Watanabe, Shota Yokoyama, Akane Kikuchi, Akihiro Shitara, Syuji Chiba, Fumiko Saito, Akihiro Sugita, Kazunari Sato, Akihiro Karube

**Affiliations:** 1Department of Obstetrics and Gynecology, Yuri-Kumiai General Hospital, Akita 015-8511, Japan; poco1031@gmail.com (R.W.); shota.yokoyama@gmail.com (S.Y.); shitaraa@med.akita-u.ac.jp (A.S.); fumiko3110@yuri-hospital.honjo.akita.jp (F.S.); akarube@yuri-hospital.honjo.akita.jp (A.K.); 2Department of Urology, Akita University School of Medicine, Akita 010-8543, Japan; naritashintaro@gmail.com; 3Department of Pathology, Yuri-Kumiai General Hospital, Akita 015-8511, Japan; misa@yuri-hospital.honjo.akita.jp (M.Y.); sugita@yuri-hospital.honjo.akita.jp (A.S.); 4Department of Urology, Yuri-Kumiai General Hospital, Akita 015-8511, Japan; a.kikuchi9@gmail.com (A.K.); chibasyuji@gmail.com (S.C.); satok@air.ocn.ne.jp (K.S.)

**Keywords:** giant hydronephrosis, ureteropelvic junction obstruction, pregnancy, congenital anomalies of the kidney and urinary tract

## Abstract

Although giant hydronephrosis (GH) associated with ureteropelvic junction obstruction (UPJO) is extremely rarely detected in pregnant women, diagnostic methods, therapeutic approaches, and perinatal management have not been established. A 31-year-old Japanese primipara had a 15 cm × 12 cm multi-cystic mass in the right abdomen detected by transabdominal ultrasound at gestational week 26. Magnetic resonance imaging revealed that the mass was right renal GH. She underwent serial ultrasound-guided transretroperitoneal drainage as conservative treatment. She delivered vaginally at gestational week 36. Since she had flank pain and a documented non-functional right kidney, laparoscopic nephrectomy was conducted 22 months after delivery. UPJO with fewer smooth muscle cells and fibrosis was histologically diagnosed in the surgical specimen. Her postpartum and postoperative courses were uneventful for 10 months. We performed a literature review of diagnostic methods, clinical characteristics, and perinatal management in pregnant women with GH due to UPJO.

## 1. Introduction

Congenital anomalies of the kidney and urinary tract (CAKUT) occur in approximately 5 per 1000 births [1]. It is a condition in which the renal pelvis, calyx, and urinary tract are congenitally dilated. The most common etiology is ureteropelvic junction obstruction (UPJO), followed by vesicoureteral reflux and ureterovesical junction obstruction [2]. UPJO is caused by intrinsic passage disorders, such as ureteral stenosis and polyps, and extrinsic passage disorders due to crossed blood vessels [3]. Intrinsic passage obstruction with physical and functional abnormalities is most common in asymptomatic UPJO [4]. Recently, the majority of patients with CAKUT in developed countries have been diagnosed with fetal screening. Patients undergo close follow-up after birth [5]. In cases not diagnosed with screening, CAKUT is diagnosed in infancy and childhood in symptomatic individuals with flank pain and urinary tract infection [6]. Although it is extremely rare, UPJO might be detected in adulthood as giant hydronephrosis (GH) in some asymptomatic patients during a medical examination or pregnancy [7,8,9,10,11,12,13,14].

We report a case of a pregnant Japanese woman with GH who was able to deliver vaginally after serial ultrasound-guided transretroperitoneal drainage. She underwent laparoscopic nephrectomy 22 months after delivery because of continuous flank pain. UPJO was diagnosed based on comprehensive macroscopic and pathological findings. We conducted a review of the English literature regarding clinical features, diagnosis, appropriate interventions, and perinatal management of pregnant women with GH due to UPJO.

## 2. Case Presentation

A 31-year-old Japanese primigravida visited our department with a complaint of secondary amenorrhea. She had no medical history and no family history, including hereditary malignancies and renal or urinary system anomalies. She was confirmed to have an intrauterine pregnancy and an intramural uterine fibroid up to 10 cm in size via transvaginal ultrasonography. The expected date of delivery was determined based on the last menstrual period. Transabdominal ultrasonography at gestational week 26 showed a right 15 cm × 12 cm multi-cystic mass, which was initially considered to be ovarian tumor by a physician. Magnetic resonance imaging (MRI) revealed that the mass was GH with thinning of the renal parenchyma, and with involvement of five vertebral levels (Figure 1A). The 10 cm fibroid was also observed (Figure 1B). She had right flank pain and irregular uterine contractions at gestational week 28. She was admitted to our hospital for those symptoms. Serum creatinine levels were normal (0.65 mg/dL) on admission.

In consultation with the urology team, ultrasound-guided aspiration was performed for symptom relief and reduction of uterine contractions. She was placed in the left lateral position. The right upper calyx was punctured using a 17 G Venuera^®^ needle; 950 mL of fluid was drained, and symptoms decreased. Drainage was performed again at gestational weeks 34 and 36, with removal of 1350 mL and 1150 mL of clear yellowish fluid, respectively.

She had premature rupture of membranes at gestational week 36 and gradually developed natural labor pains. Due to non-reassuring fetal status, she underwent vacuum extraction and gave birth to a healthy male newborn weighing 2236 g with Apgar scores of 8/9. Her puerperal course was uneventful. UPJO with GH was followed by the urology team. At 20 months after delivery, she had a relapse of intermittent right flank pain. Three-dimensional computed tomography (CT) showed that the right renal cortex was slightly enhanced and paper-thin. A Tc99m dimercaptosuccinic acid (DMSA) renal scan and 24 creatinine clearance from the right renal nephrostomy (6.9 mL/min) revealed no right kidney function (Figure 2A,B). She underwent laparoscopic right nephrectomy for pain relief (Appendix A). Macroscopically, the right renal parenchyma was thin and obstruction at the ureteropelvic junction was confirmed (Figure 3A–C). Ureteral stenosis with fewer smooth muscle cells and fibrosis were histologically confirmed. The diagnosis was UPJO (Figure 4A–C). Her postpartum and postoperative courses were uneventful for 10 months after surgery.

## 3. Discussion

UPJO is the most common cause of GH in adulthood. The majority of patients have CAKUT detected during fetal screening or a symptomatic childhood [5]. Some asymptomatic patients who are missed by screening can be diagnosed with GH in adulthood during a medical examination or pregnancy, as the disease progresses slowly [7,8,9,10,11,12,13,14]. UPJO in adults is more common in women and on the left side [15,16,17]. By contrast, hydronephrosis associated with normal pregnancy is more common on the right side due to compression of the right ureter with rotation of the uterus caused by the sigmoid colon [18]. The etiology of UPJO has been reported to include genetic alternations and maternal environmental factors, such as fetal exposure to non-steroidal anti-inflammatory drugs, diabetes mellitus, and angiotensin-converting enzyme inhibitors [19,20]. However, the distinct pathogenesis is largely unknown.

We conducted a literature review of pregnancy with GH due to UPJO. We searched the PubMed database using the keywords “hydronephrosis” and “pregnancy.” A total of eight patients, including our patient, were identified (Table 1) [8,9,10,11,12,13,14]. All patients were primiparas, and five involved the left kidney. GH was mostly observed during the second trimester with transabdominal ultrasound. Two cases showed the mild to moderate increase in serum creatinine. Symptoms were alleviated by conservative therapies such as aspiration and urinary stenting. With maternal and fetal adaptation, five patients delivered vaginally and two patients underwent cesarean section. Since pregnant women with UPJO are extremely rare, there are no consensus-based management guidelines. According to previous reports, invasive nephrectomy was performed during pregnancy [11,13,14]. However, in recent years, less invasive management, including ureteral stenting and drainage, has been preferred due to advances in imaging and the desire to avoid fetal anesthesia and perioperative maternal infections [9,12]. In our patient, serial transretroperitoneal drainage was performed to relieve flank pain because she refused an indwelling catheter and stent. Since serial aspiration constitutes temporary management and is associated with a high probability of complications, indications of the aspiration should be carefully considered. We speculated that drainage secured a space in the abdominal cavity to prolong the gestational period by releasing direct pressure on the uterus. As shown in previous reports, the method of delivery is usually the same as in healthy pregnant women, and is based on maternal and fetal adaptation. In addition, uterine fundal pressure should be avoided as it carries the risk of GH rupture [21].

There are some suggestions for early detection of UPJO during pregnancy. While serum creatinine has increased approximately 40% (8/19) in adult patients with GH [7], serum creatinine is not included in routine blood tests recommended for pregnant women in Europe and the United States [22,23]. The initial antenatal checkup is the first opportunity for a blood test in healthy women who have never had regular checkups, such as our patient. Serum creatinine blood tests might be considered as an assessment of baseline renal function, especially in primiparas, not only for UPJO, but also for hypertensive disorders of pregnancy, hemolysis, elevated liver enzymes, low platelet count syndrome, and acute fatty liver of pregnancy, conditions in which renal function decreases during the perinatal period [24]. UPJO was detected during the second trimester in most previously reported cases. This indicates that it is difficult to detect GH with transvaginal ultrasound during the first trimester. However, GH could be detected with transabdominal ultrasound performed during the second trimester as the uterus grows. Therefore, minimally invasive and convenient transabdominal ultrasound during the first trimester is the most suitable method for early detection of UPJO, in particular for primiparas with abdominal or back pain and uncertain renal dysfunction. MRI and CT should be used for definitive diagnosis of GH due to UPJO.

## 4. Conclusions

We experienced a case of a pregnant woman with GH due to UPJO who was able to deliver vaginally with serial ultrasound-guided transretroperitoneal drainage of urine from the GH. She was diagnosed with UPJO based on radiographic, macroscopic, and pathological findings. Although it is difficult to establish a consensus-based management guideline for UPJO in pregnancy, conservative management is considered to be a suitable option for symptom relief and maintaining the pregnancy. In general, nephrectomy is considered to be performed after delivery, unless the failure of conservative management and severe complication due to GH are observed. The method of delivery without uterine fundal pressure should be determined according to maternal and fetal adaptation. Obstetricians and gynecologists need to pay particular attention to pregnant women who develop symptoms from genitourinary disorders during the perinatal period.

## Figures and Tables

**Figure 1 diagnostics-12-00913-f001:**
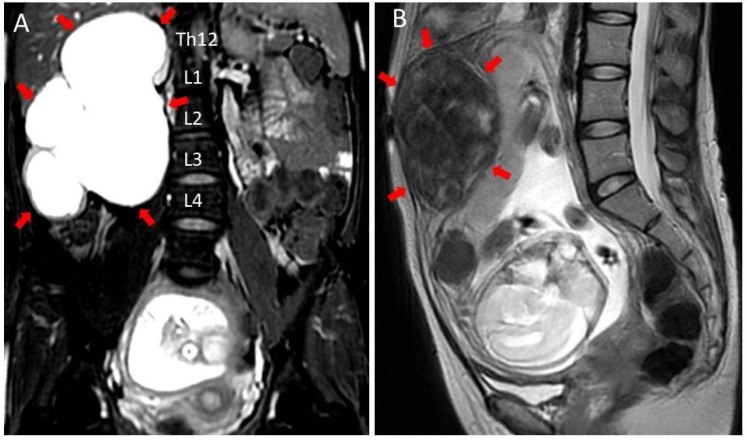
Magnetic resonance imaging shows a thin and multilocular right kidney measuring 15 cm × 12 cm in size (red arrows), approximately the height of five vertebral bodies. (**A**) coronal view. Uterine fibroids with partial degeneration (10 cm × 5 cm/red arrows) were found in the anterior uterine wall (**B**) sagittal view.

**Figure 2 diagnostics-12-00913-f002:**
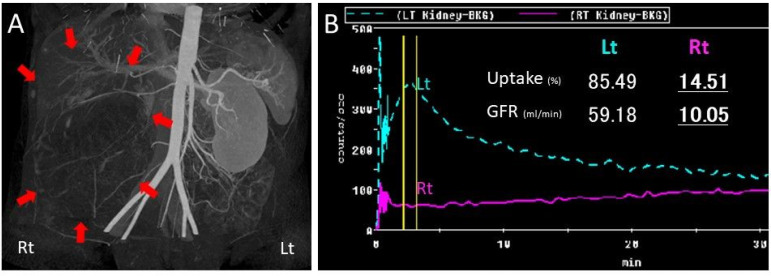
Three-dimensional computed tomography showed that the cortex of the right kidney indicated by the red arrow was slightly enhanced (**A**). Tc99m dimercaptosuccinic acid renal scan revealed the uptake and glomerular filtration rate (GFR) was 85.49% and 59.18 mL/min from the left kidney and 14.51% and 10.05 mL/min from the right kidney, respectively (**B**).

**Figure 3 diagnostics-12-00913-f003:**
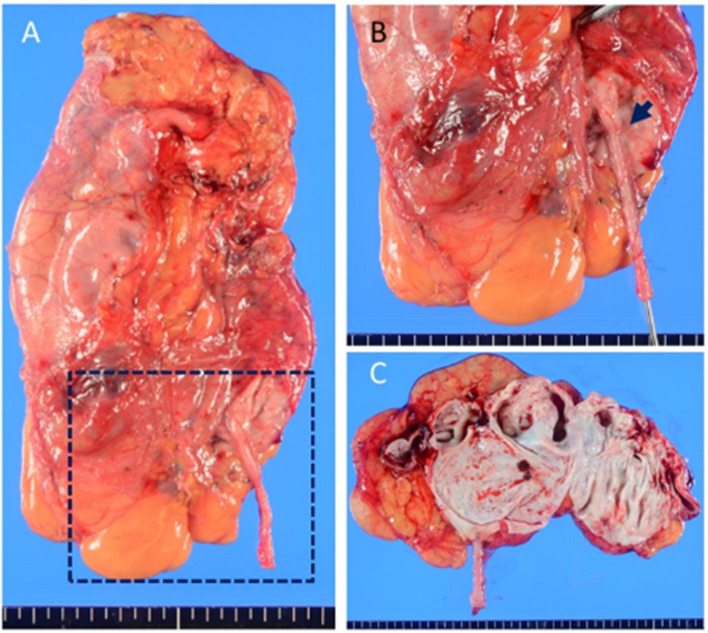
Gross image of the right kidney specimen (**A**). A higher magnification image of the dotted rectangle in (**A**) shows the site of right ureteral stenosis indicated by the blue arrow (**B**). The lumen of the cortex of the right kidney was paper-thin and multilocular (**C**).

**Figure 4 diagnostics-12-00913-f004:**
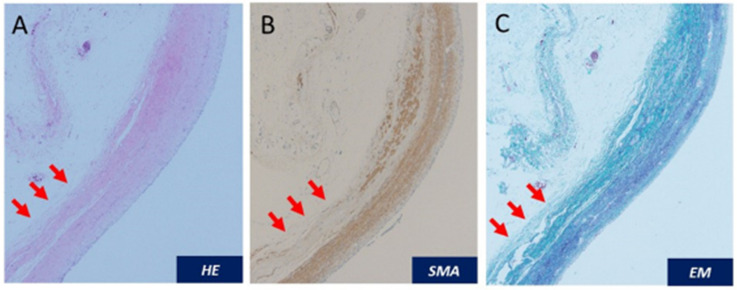
Hematoxylin and eosin (HE) staining at the site of right ureteral stenosis (**A**). The middle circular muscle layer (red arrows) had disruption of smooth muscle cells shown by smooth muscle actin (SMA) staining (**B**) and fibrosis shown by Elastica Masson (EM) staining (**C**).

**Table 1 diagnostics-12-00913-t001:** Literature review of pregnancy with GH due to UPJO.

First Author	Age	Parity	Family History of Urinary Abnomality	GH Site	Symtom	Diagnostic Method	Timing of GH Detection	Serum Creatinine (mg/dL)	Pregnancy Complication	Treatment during Pregnancy	Delivery Method/Week
Present case (2022)	31	0	None	Right	Flank pain	TAU/MRI	26th	0.65	None	Aspiration (3 times)	Vaginal delivery/36th
Ramly F [8] (2021)	27	0	None	Left	Iliac fossa pain	TAU/MRI	23rd	Normal	GDM/HDP	None	C/S due to pre-eclampsia/32th
Nerli RB [9] (2016)	25	NA	NA	Left	Colicy pain	TAU/MRI	24th	1.4	None	Double J stenting	NA
Lin YJ [10] (2013)	31	0	NA	Left	None	Ultrasound examination /MRI	28th	0.7	None	None	C/S due to cephalopelvic disproportion/term
Mastoroudes H [11] (2007)	32	0	NA	Left	Sided-pain	Ultrasound scan /Intravenous urograph	Before pregnancy	Normal	None	Nephrectomy	Vaginal delivery/term
Peng HH [12] (2003)	21	0	NA	Left	Flank pain	TAU/MRI	25th	0.9	None	Aspiration (3 times)	Vaginal delivery/39th
Bernstine RL [13] (1959)	19	NA	NA	Right	Dull aching sensation	Palpation /Intravenous urogram	26th	Normal	None	Nephrectomy	Vaginal delivery/term
Hecht EL [14] (1952)	17	0	None	Right	None	Palpation	31st	NA	None	Nephrectomy	Vaginal delivery/34th

GH, giant hydronephrosis; NA, not available; TAU, transabdominal ultrasonography; MRI, magnetic resonance imaging; GDM, gestational diabetes mellitus; C/S, cesarean section; HDP, hypertensive disorders of pregnancy.

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
