# Peer review of "Perinatal Management in a Pregnant Woman with Ureteropelvic Junction Obstruction: Case Report and Literature Review"

_diagnostics, 2022, doi:10.3390/diagnostics12040913_

Round 1

Reviewer 1 Report

Dear Authors,

Your case report is interesting.

In my opinion small corrections are needed.

  1. "threatened abortion" I am not sure if this words are used properly, in more cases  pregnancy after 22 hbd is preterm delivery.
  2. Conclusions should be written as separated part.

Author Response

Reviewer #1 

  1. Threatened abortion

Response: We deleted the term "Threatened abortion", and added the sentence in the revised manuscript. 

  1. Conclusions

Response: We corrected "Conclusion paragraph" as a separate part. 

Reviewer 2 Report

The authors reported a rare case of perinatal management for a pregnant woman with ureteropelvic junction obstruction. This case report will be useful for the future management to help pregnant women who are suffering from ureteropelvic junction obstruction. Introduction and case presentation are described properly. I have only minor comments for discussion section and hope they improve this report.

Minor comments:

Discussion, page 3: As a conclusion from this case report, conservative management was suggested as a better option than invasive nephrectomy, which was performed at least until 2007 in literatures. From the diagnostic point of view, I understand it is difficult to make a guideline, but is there any suggestive criteria to decide or consider for medical teams which treatment they should conduct for pregnant women? Conservative management will be preferred, but when the teams need to think about nephrectomy during gestation? I leave the authors to decide if they add sentences to describe their suggestive criteria.

Figure 1: Although multilocular right kidney and uterine fibroids are obvious in Figure A and B, I suggest pointing them by coloured arrows like the other figures.

Author Response

Reviewer #2 

  1. Suggestive criteria as the conclusion

Response: We added and described the sentence with nephrectomy in the revised manuscript. 

  1. Figure 1

Response: We added red arrows in the Figure 1A and 1B.